# Molecular Pathology of Skin Melanoma: Epidemiology, Differential Diagnostics, Prognosis and Therapy Prediction

**DOI:** 10.3390/ijms23105384

**Published:** 2022-05-11

**Authors:** József Tímár, Andrea Ladányi

**Affiliations:** 12nd Department of Pathology, Semmelweis University, 1191 Budapest, Hungary; 2Department of Surgical and Molecular Pathology and the National Tumor Biology Laboratory, National Institute of Oncology, 1122 Budapest, Hungary; ladanyi.andrea@oncol.hu

**Keywords:** skin melanoma, genomics, molecular pathology, prognostic and predictive markers

## Abstract

Similar to other malignancies, TCGA network efforts identified the detailed genomic picture of skin melanoma, laying down the basis of molecular classification. On the other hand, genome-wide association studies discovered the genetic background of the hereditary melanomas and the susceptibility genes. These genetic studies helped to fine-tune the differential diagnostics of malignant melanocytic lesions, using either FISH tests or the myPath gene expression signature. Although the original genomic studies on skin melanoma were mostly based on primary tumors, data started to accumulate on the genetic diversity of the progressing disease. The prognostication of skin melanoma is still based on staging but can be completed with gene expression analysis (DecisionDx). Meanwhile, this genetic knowledge base of skin melanoma did not turn to the expected wide array of target therapies, except the BRAF inhibitors. The major breakthrough of melanoma therapy was the introduction of immune checkpoint inhibitors, which showed outstanding efficacy in skin melanoma, probably due to their high immunogenicity. Unfortunately, beyond *BRAF*, *KIT* mutations and tumor mutation burden, no clinically validated predictive markers exist in melanoma, although several promising biomarkers have been described, such as the expression of immune-related genes or mutations in the IFN-signaling pathway. After the initial success of either target or immunotherapies, sooner or later, relapses occur in the majority of patients, due to various induced genetic alterations, the diagnosis of which could be developed to novel predictive genetic markers.

## 1. Introduction

Pathological diagnostics of cutaneous melanoma was established in the past decades and mostly based on histopathological characteristics completed with a relatively simple immunohistochemical marker set. However, this situation profoundly changed recently: the widespread dermatological melanoma screening programs detect premalignant lesions with a much higher frequency, which requires highly sensitive molecular tests to prove malignancy. On the other hand, the genetic background of hereditary melanoma became more and more complex, again defining the need for more complex genomic testing. Last but not least, since cutaneous melanoma is the most metastatic human cancer but detected at earlier stages, there is a clinical need of more precise prognostication, which can be based on newly developed genetic tests. Meanwhile, the identification of driver genes of skin melanoma helped to develop target therapies for what predictive genetic characterization became an everyday practice. However, the most effective therapy of skin melanoma is immunotherapy; unfortunately, its predictive markers have not yet entered clinical practice, although our knowledge of the immunogenomic characteristics of skin melanoma has increased enormously in the past years. In this review, we briefly summarize our basic knowledge on skin melanoma genomics, with the aim to show its clinicopathological relevance and highlight those areas where, although we have the required genetic knowledge, its introduction into clinical practice is urgently needed.

## 2. Molecular Epidemiology

Malignant melanoma is one of the most metastatic human cancers where a T1 sub-millimeter sized primary tumor of a ~10^6^ cell population can have a significant metastatic potential, compared to most solid cancers, where a ten-times larger but similar T1 tumor of a population of 10^9^ cells may not have it. Due to the novel lifestyles and global atmospheric changes, UV exposure of the skin has increased gradually, resulting in a paralleled increase in the incidence of melanoma [1]. In most European countries, melanoma can be found among the ten most frequent malignancies [1], and its prominent metastatic potential presents a significant burden for healthcare providers.

Malignant melanoma can develop from benign nevi or de novo. Considering the high incidence of benign nevi, the malignant transformation potential of these lesions is fortunately low. Meanwhile, nevi carry, at high frequency, the signature UV-induced mutation of *BRAF* (v-Raf murine sarcoma viral oncogene homolog B1) at exon 15, providing evidence of the etiological factor behind [2]. Malignant melanoma, however, can develop on non-UV-exposed skin, mucosal epithelium or uvea, and these melanoma types usually lack the characteristic *BRAF* mutation.

Both skin and uveal melanoma can have familial form, but their genetic background is different. Besides the loss of *CDKN2A* (cyclin-dependent kinase inhibitor 2A), germline mutations of *CDK4* (cyclin-dependent kinase 4), *MITF* (microphtalmia-associated transcription factor) and *BAP1* (BRCA1-associated protein 1) are the most significant contributors for hereditary melanoma [3]. However, the picture became more complex with the discovery of germline alterations of the pigmentation-related and DNA-repair-related genes in the development of melanoma. As far as the pigmentation-related genetic factors are concerned, besides *MITF* mutations, the alterations of MITF-regulated *MC1R* (melanocortin-1 receptor), *SLC45A2* (solute carrier family 45 member 2) and *OCA2* (oculocutaneous albinism type 2) genes, as well as those of the melanosomal *TYR* (tyrosinase) and *TYRP1* (tyrosinase-related protein 1) and DNA repair gene defects of *TERT* (telomerase reverse transcriptase) and *APEX1* (apurinic/apyrimidinic endodeoxyribonuclease 1), are also significant contributors, increasing the risk of melanoma development. Furthermore, novel germline alterations at the chromosomal region of 1q21.3 involving *ARNT* (aryl hydrocarbon receptor nuclear translocator) and *SETDB1* (SET domain bifurcated histone lysine methyltransferase 1) were discovered lately as possible genetic risk factors for melanoma [3,4,5,6].

It is worth mentioning that, in the case of uveal melanoma, inherited homologous recombination or mismatch repair deficiencies due to *PALB2* (partner and localizer of BRCA2) or *MLH1* (MutL homolog 1) are the primary causes for heritability [7].

## 3. Molecular Classification

The sequencing of thousands of malignant melanomas worldwide defined the atlas of genomics of melanoma [8]. These analyses revealed that the most frequent gene defect of skin melanoma is the activating mutation of *BRAF* oncogene in exon 15/codon 600, characterizing almost half of these tumors. At a significantly lower frequency (~20%), the *NRAS* (neuroblastoma RAS viral (*v-ras*) oncogene homolog) oncogene is mutated in melanoma in exon 3/codon 61. Interestingly, almost with similar frequency (<15%) the *KIT* (*KIT* proto-oncogene, receptor tyrosine kinase) gene is also mutated in melanoma [9] (Figure 1). It has to be emphasized that the KIT receptor signaling pathway, containing NRAS and BRAF, is the dominating pathway in melanocytes (and evidently in melanoma), and it is responsible for activating the melanocyte-specific transcription factor MITF. As in other cancers, several oncosuppressor genes are mutated in melanoma, including *TP53* (tumor protein p53), *NF1* (neurofibromin 1), *CDKN2A* and *PTEN* (phosphatase and tensin homolog), at a similar relatively low frequency (~15%) (Figure 1). However, there are also genome-wide copy number alterations in melanoma: amplification affects the melanoma oncogenes, as well as *CCND1* (cyclin D1) and *MITF*, while loss of heterozygosity (LOH) or complete loss may affect *CDKN2A* (p16) and *PTEN* [1]. Moreover, at much lower frequencies, chromosomal rearrangements affecting (beside *PTEN*) the kinase receptors *ALK* (anaplastic lymphoma kinase), *RET* (ret proto-oncogene) and *NTRK* (neurotrophic tyrosine receptor kinase) can also be detected [2].

Similar to the traditional histological subclassification of melanoma, today the molecular classification is also possible where there are four major categories, the *BRAF*-mutant, the *RAS*-mutant, the *NF1*-mutant and the so-called triple wild-type forms (Table 1) [8]. It is now evident that a cancer is characterized by a relatively well-described set of driver oncogenes. Accordingly, the *BRAF*-mutant melanoma is a p16-lost or negative tumor, where *TP53* mutations are relatively rare, but this is the form where the *MITF* and *PD-L1* (programmed death ligand 1) genes are amplified. *RAS*-mutant melanomas are different from *BRAF*-mutant ones because neither *MITF* nor *PD-L1* are amplified, but *TP53* is more frequently mutated. The *NF1*-mutant melanoma can be called a suppressor gene melanoma, since, besides *NF1*, *CDKN2A*, *RB1* (retinoblastoma 1) and *TP53* genes are all mutant. Last but not least, the so-called triple wild-type melanomas are *TP53*-wild-type but carry mutations of *MDM2* (mouse double minute 2 homolog) and *CCND1* [10]. All four molecular subtypes are characterized by *IDH1* (isocitrate dehydrogenase 1) mutation involved in epigenetic regulation, while *ARID2* (AT-rich interaction domain 2) is wild type only in the triple wild-type form but mutated in the other subclasses, resulting in disturbances in chromatin remodeling and transcriptional control. It is another difference that the AURKA (Aurora kinase A) inhibitor *PPP6C* (protein phosphatase 6 catalytic subunit) gene is mutated in *BRAF*- and *RAS*-mutant subclasses exclusively.

The most frequent histological variant of skin melanoma is the superficial spreading melanoma (SSM) type. Other frequent variants are nodular melanoma (NM), acral lentiginous melanoma (ALM) and lentigo maligna melanoma (LMM). It is interesting that, in SSM or NM histological forms, the mutation order is *BRAF* > *NRAS* > *KIT*, while in the (acral-)lentiginous forms, the order of oncogene mutation frequency is *KIT* > *BRAF* > *NRAS*. Furthermore, ALM is characterized by chromosomal instability and a low mutational burden. There are rare histological variants of melanoma, with unique molecular signatures. The driver oncogene of deep penetrating melanoma is *GRIN2A* (N-methyl-D-aspartate receptor glutamate ionotropic receptor NMDA type subunit 2A), while the nevus-like melanoma is characterized by mutations of the lipid/AKT signaling pathway. A rare histological variant is the desmoplastic melanoma arising on chronic sun-damaged skin and is uniquely characterized by *NFKBIE* (NFKB inhibitor epsilon) promoter mutation, rare types of *BRAF* mutation and high tumor mutational burden (TMB) [11]. The blue nevus melanoma is a prototype of *CDKN2A*-lost tumor. As compared to these variants of (skin) melanoma, uveal melanoma is characterized by genetic alterations of the melanocortin receptor-1 signaling due to the mutations of *GNAQ* and *GNA11* (guanine nucleotide-binding protein alpha subunit q and alpha subunit 11) genes [2].

## 4. Molecular Diagnostics

The identification of melanocytic lesions is based on specific markers of melanocytes which all associate with melanosomes not expressed by any other cell linages. Maturation of melanosomes is a four-step process, where lipid membranes of this organelle begin to contain melanosome-specific protein Pmel17/gp100, after which tyrosinase enzyme will be synthesized later, together with dopachrome tautomerase enzyme, and ultimately the organelle will contain MART-1/MelanA [12]. Based on this, the identification of melanocytic cells can be performed by immunohistochemistry detecting melanosomal proteins Pmel17/gp100, MART-1, or tyrosinase. Since the transcription of these genes is controlled by melanocytic MITF and SOX10 (Sry-related HMg-Box gene 10), the immunohistochemical detection of these transcription factors can also be used as a specific melanocytic marker. There is also a widely used, less specific protein marker of melanocytes, S100B (S100 calcium binding protein), which is expressed by neural cells as well (Table 2) [13].

Meanwhile, the diagnostic problem is frequently not the melanocytic origin of the lesion but the potential malignancy. Histopathology is the gold standard of differentiating these lesions, and the MPATHDx classification and its appropriate interpretation could help [14]. Immunohistochemical detection of the nuclear protein Ki67 is not suitable for this distinction since nevi, especially those mechanically damaged, may contain proliferating nevocytes. Until recently, morphological analysis of the melanocytic tumor cells served as the only diagnostic help, but today there are genetic techniques which could help in objectively defining the nature of the melanocytic lesions. One possibility is to use immunohistochemical markers of malignancy: two such markers have been evaluated and validated, p16 and PRAME. Loss of p16 protein alone may not be an optimal tool to differentiate benign or malignant lesions, but its combination with Ki67 and Pmel/gp100 may better suit the diagnostic need [14,15]. A new alternative to p16 is high PRAME protein expression, which has been validated relatively extensively [14]. Another possibility is to use a four-gene fluorescence in situ hybridization (FISH) probe applicable to formalin-fixed paraffin-embedded (FFPE) blocks. This probe set is composed of genes which characteristically suffer from copy number variations during malignant transformation of melanocytes: gene amplification generally occurs in *RREB1* (rat responsive element binding protein 1) and *CCND1* genes, while the loss of copies occurs in the case of *CDKN2A* and *MYB* (*MYB* proto-oncogene and transcription factor) genes (Table 3). A minimum of three copy number variations of these genes is required for malignancy definition [16]. Recently a gene expression signature was defined for melanoma, which could be applied to FFPE sections, as well, to discriminate melanomas from non-malignant melanocytic lesions. This molecular test (myPath; Myriad) is based on RNA evaluation of 14 genes, 7 of which are melanoma genes and 7 are tumor-microenvironment-associated ones (Table 3) [17].

## 5. Immunological Characteristics—The Tumor Immune Microenvironment

Skin melanoma is traditionally considered one of the most immunogenic tumor types, based in part on its long-known feature of frequently containing a characteristic lymphoid infiltrate; furthermore, it may be the only tumor type for which spontaneous regression can occur in the primary tumor; this regression is assumed to be the consequence of antitumor immune response [18]. More recent research and therapy results supported the unique immunological features of cutaneous melanoma from other aspects. It belongs to tumors with the highest tumor mutational burden (TMB), caused by high mutagen exposure (UV radiation) [19,20,21]. As a consequence of high mutation rate, the chance of production of mutant proteins that may function as neoantigens is increased, contributing to enhanced immunogenicity [19,22]. This presumably explains the outstanding efficiency of antitumor immunotherapies, including immune checkpoint inhibitors (ICIs), in melanoma patients. Studies applying different gene panels characterizing local immune activity, based on The Cancer Genome Atlas (TCGA), also indicate high immune activity in melanomas [23,24]. The above features, however, do not apply to the rarer melanoma types, such as acral types, mucosal types, or to uveal melanomas; in these, both the mutational burden and immune-associated gene expression generally show lower values, and, consequently, their immunotherapy sensitivity is lower [23,24,25,26].

Since cutaneous melanoma belongs to tumors with high mutation frequency, a further increase in TMB and the amount of neoantigens during progression is not expected, although an increase in defects of homologous recombination repair can be detected [19]. However, from the point of antigen presentation, genetic defects might abrogate the beneficial effect of high neoantigen burden. In melanoma, LOH or mutations in the chromosomal regions, where genes encoding human leukocyte antigen (HLA) class I heavy chains (chromosome 6) or beta2-microglobulin (B2M, chromosome 15) are located, are relatively frequent [19], and they may render such tumors immunoresistant because of the crucial role of HLA class I molecules in antigen presentation to CD8^+^ cytotoxic T lymphocytes. It is also important that the *BRAF*-mutant melanomas may also develop *PD-L1* gene amplification [10], which may result in immunoresistance, although it could increase sensitivity to ICIs targeting the PD-1/PD-L1 axis.

## 6. The Molecular Background of Melanoma Progression

One of the outstanding questions in melanoma progression is how stable the oncogenic drivers are. Most of our data come from investigating primary tumors or locoregional metastases, while very few genomic data are available concerning visceral metastases. It is known that melanoma is also a clonally heterogeneous tumor where driver mutant and wild-type clones are present as a mixture in the primary. We have analyzed the driver gene presence and the ratio of the mutant clones in melanoma metastases as compared to the primary tumors. We did not observe a complete loss of the driver oncogenes (*BRAF* or *NRAS*) in visceral metastases. However, we have found an extreme heterogeneity concerning the relative clonal ratio of the driver clones, since we found genetic evidence for all the three possible scenarios: maintenance of the original ratio, significant decrease of the driver clones and significant increase of the driver clones in metastases [27]. Accordingly, based on the clonal dominance of a driver clone in the primary tumor (or their extreme subclonality), one cannot predict the situation in the metastases which can be important when indicating target therapies.

The natural genetic progression of melanoma without therapeutic pressure is an important process. The data indicate that there are several novel mutations which emerge in metastases, such as those of *BRCA1* (breast cancer gene 1), *EGFR4* (epidermal growth factor receptor 4) and *NMDAR2* (N-methyl-D-aspartate receptor 2). Since *BRCA1* mutation results in homologous recombination deficiency, it may open the way to explore the potential use of PARP inhibitors in those instances. Furthermore, copy number changes are also emerging, affecting *MITF* or *MET* (*MET* proto-oncogene and receptor tyrosine kinase) (amplifications), or the loss of the suppressor *PTEN*, increasing the genetic diversity of the metastases as compared to the primary tumor [2,10]. Furthermore, copy number variations developing in metastasis-associated genes, *NEDD9* (neural precursor cell expressed, developmentally downregulated 9), *TWIST1* (Twist family BHLH transcription factor 1), *SNAI1* (Snail family transcriptional repressor 1) and *TEAD* (transcriptional enhanced associate domain) are also significant genetic contributors of progression [2,10]. A recent study focusing on the genetic analysis of visceral metastases of melanoma revealed organ-specific genetic alterations of progression. In the case of lung metastasis, copy number gains have been observed in several (19) immunogenic genes—most of them found to be expressed at protein levels (13)—termed as immunogenic mimicry, indicating a strong immunologic selection mechanism operational in this form of metastasis [28]. This observation may suggest that visceral metastases may not be equally sensitive to immunotherapy. In contrast to lung metastasis, in brain metastases of melanoma, besides *TERT*, amplifications of *HGF* (hepatocyte growth factor) and *MET* genes have been found, indicating the presence of a possible autocrine loop of signaling, offering a potential target therapy option for this type of metastasis. As compared to these organs, liver metastases did not contain many unique genetic alterations, except for amplifications of *CDK6* (cyclin-dependent kinase 6) and *MAPK* (mitogen-activated protein kinase) genes; both can now be targeted by clinically tested drugs [28]. Collectively, these genetic data offer new possibilities for target therapies of progressing melanoma and hopefully would initiate new types of clinical trials.

## 7. Prognostic Markers: Gene Expression Pattern

Primary skin melanoma can be classified into three molecular categories based on gene expression signature: a proliferative one driven by the SOX10–MITF pathway, where *CDKN2A* is lost; an invasive one characterized by activity of epithelial–mesenchymal transition genes *SNAI1*, *ZEB1* (zinc finger E-box binding homeobox 1) and *TGFBR2* (transforming growth factor beta receptor 2); and a so-called immune-mediated one characterized by activation of the tumor microenvironment [8,10]. Other analyses mostly confirmed these genotypes, defining the *MITF* low/proliferative, the high-immune response, the *MITF* high/pigmentation and the normal subclasses [11] where the *MITF* low subclass has the poorest prognosis. Lately, the TCGA analysis of the melanoma gene expression signatures also found the *MITF* low group, the immune group and the keratin groups [11]. These genetic signatures also define the characteristic metabolic profiles of melanoma. The proliferative phenotype is characterized by a high expression of *PGC1A* (peroxisome proliferator γ coactivator α), which is responsible for the production of reactive oxygen species (ROS), the high level of which results in chemoresistance, as well as immunoresistance [29]. Furthermore, the TCGA melanoma database analysis identified a prognostic signature containing *CAV1* (caveolin 1), *CD36* and *CPT1C* (carnitine palmitoyltransferase 1C), members of which are responsible for fatty acid uptake and metabolism [30]. Unfortunately, none of these signatures has been analyzed in prospective clinical trials for their prognostic power. As single prognostic markers, *EGFR4* overexpression was found to have a negative prognostic impact, while *ALDH1* (aldehyde dehydrogenase 1) overexpression has a positive prognostic impact [31]. Furthermore, traditional serum biomarkers such as 5-S-cisteinyldopa or lactate dehydrogenase (LDH) may also serve as prognostic factors; a high expression of them characterizes poor prognosis [31]. LDH overexpression is the result of the increased glycolytic activity and is released from hypoxic and necrotic melanoma cells into the circulation. The continuous monitoring of LDH levels upon target therapy or immunotherapy can also be used to assess efficacy since responding patients are characterized by normalizing LDH levels [32].

A recent development is the application of circulating DNA tests as possible prognostic factors: several studies provided evidence for the prognostic power of *BRAF* and *NRAS* mutations detected in the peripheral blood as markers of poor prognosis, defining molecular residual disease [33]. However, the most extensively validated prognostic melanoma gene signature is a 31-gene expression panel constructed from meta-analyses of skin and uveal melanoma signatures, containing four inner control genes. This 31-gene panel was retrospectively and prospectively analyzed clinically and proved to have strong independent prognostic power (Table 4) [34].

## 8. Prognostic Markers: The Tumor Immune Microenvironment

Based on the favorable immunological features described above, one could expect that the density or composition of tumor-infiltrating immune cells will have a prognostic role as well. However, there is no clear association between the amount of tumor-infiltrating immune cells and TMB or neoantigen burden, and the intensity of infiltration by immune cells (e.g., CD8^+^ T cells, among others) in melanoma is not outstanding compared with other tumor types [19,35,36,37].

The prognostic value of immune cell infiltration in primary melanoma was analyzed in many studies. In the earliest investigations, the number of tumor-infiltrating lymphocytes (TILs) was determined based on hematoxylin–eosin staining; although a prominent lymphocyte infiltration (especially if it was determined in the vertical growth phase) proved to be a significant independent parameter of longer survival in several studies, in other cases, no such association was found, or the independent prognostic role was not confirmed [38,39]. In studies based on the immunohistochemical detection of immune cell type specific markers, controversial results were reported on the prognostic value of infiltration by T lymphocytes (including CD4^+^ and CD8^+^ subsets), or macrophages. On the other hand, a favorable prognostic effect of mature dendritic cells, as well as B cells, was described [38].

In studies evaluating immune cell infiltration in metastases of cutaneous melanoma (focusing mainly on lymph node, subcutaneous or cutaneous metastases), the amount of total TIL was found associated with patients’ survival [35]. Few studies have analyzed the prognostic impact of specific immune cell subsets. According to those comprising the largest patient cohorts [40,41], a high density of CD3^+^/CD8^+^ T lymphocytes, as well as that of CD20^+^ B cells, predicted favorable outcome.

In the past years, numerous transcriptomic analyses (performed on either primary or metastatic melanoma tissues) yielded the classification of samples based on the expression of immune-related genes, showing an association of “high immune” sample subsets or characteristic gene signatures with favorable outcome of the disease [8,23,42,43,44]. In contrast to the above observations on cutaneous melanomas, in uveal melanomas, a high expression of immune-cell-infiltration-associated genes or high immune scores were found to correlate with lower survival rate [45,46], in accordance with findings on the association of high density of T cells and macrophages with poor prognosis in this melanoma type [47,48].

## 9. Predictive Markers of Melanoma

The therapy of advanced/metastatic melanoma is based on its molecular classification since chemotherapies are more or less ineffective and irradiation has only a limited effectivity, mainly in the case of brain metastases. As was shown, there are three major driver oncogenes defining three major molecular forms of skin melanoma, *BRAF*-, *NRAS*- and *KIT*-mutant, and there is a fourth which is the so-called triple wild-type form. Accordingly, the molecular characterization of melanoma is necessary before making therapy decisions. However, target therapies are only approved in the case of exon 15/codon 600 *BRAF*-mutant melanoma, using a BRAF inhibitor with or without MEK inhibitors [49]. At the moment, there is no approved drug for *NRAS*-mutant melanoma, although MEK inhibitors are under clinical testing with some encouraging results [50]. In the case of *KIT*-mutant melanomas, it is important to define the KIT-inhibitor sensitive ones based on experiences with KIT inhibitor efficacy in gastrointestinal stromal tumor (GIST). In the case of the triple wild-type melanoma, the only recommended therapy option is immunotherapy in the form of anti-PD-1 (programmed cell death protein 1) and/or anti-CTLA-4 (cytotoxic T cell antigen 4) antibodies [49]. It is also important that immunotherapies can also be introduced in driver gene positive melanomas. It is a question what the rationale of the therapeutic decision would be in such cases. It was mentioned earlier that the clonal composition of the melanoma for a given oncogenic driver can be variable and can be objectively determined by assessing the variant allele frequency (VAF) of the mutation. In the case of subclonality (<20% VAF), the major part of the tumor is composed of tumor cells carrying the wild-type oncogene, where the efficacy of a target therapy is questionable, and early relapse and development of resistance is expectable. On the other hand, in the case of heterozygous mutation (~50% VAF), 100% of the tumor population carries the mutant oncogene, and the chance to control the disease is high.

Meanwhile, even with target or immunotherapies, a majority of melanomas progress due to genetic progression under special environmental pressure induced by the therapy. In the case of BRAF inhibitor/MEK inhibitor therapies, novel resistance mutations were reported, affecting *BRAF* outside V600, *MEK1/2* and various members of the AKT signaling pathway (*AKT1* (AKT serine/threonine kinase 1), P*IK3CA* (phosphoinositide-3-kinase catalytic subunit alpha) and *PIK3R1/2* (phosphoinositide-3-kinase regulatory subunit 1/2)) [2,10,51]. Furthermore, gene amplifications of *BRAF* or *MITF* were also detected in target-therapy-resistant tumors. Last but not least, it seems that the loss of PTEN can also activate the AKT signaling pathway, resulting in BRAF inhibitor resistance [51].

The most widely used immunotherapies in patients with cutaneous melanoma are immune checkpoint inhibitors, monoclonal antibodies targeting PD-1 or CTLA-4. In the case of these agents, no clinically validated predictive markers exist in melanoma. PD-L1 expression of tumor cells and/or tumor-infiltrating immune cells is a prerequisite of anti-PD-1/PD-L1 treatment for several cancer types. However, there is no such requirement in the case of melanoma; although tumor-cell PD-L1 expression showed an association with therapeutic effect in several studies, responses can be observed in a significant proportion of PD-L1-negative cases, as well. Considering both the therapeutic benefit and the risk of side effects, in cases with positive tumor cell PD-L1 staining (>5%), anti-PD-1 monotherapy is advantageous, while, in negative cases, combination with anti-CTLA-4 could be more advantageous [52].

Besides PD-L1 expression, the most frequently analyzed tissue biomarker of immunotherapy efficacy is TMB, which showed a positive association with ICI therapy response [53,54]. Microsatellite instability of melanoma is rare; therefore, it cannot be efficiently used for selection for immunotherapy [2,10]. Moreover, several other potential predictive factors came to light, including clinical parameters as tumor burden or localization of metastases; blood or serum markers, e.g., absolute number or proportion of some blood cell types, or LDH level; the composition of the intestinal microbiome and microbial gene signatures; and tumor-infiltrating immune cells or expression of immune-associated genes in the tumor [39,54,55,56,57,58,59]. Loss of HLA class I expression could also contribute to both primary and acquired immunotherapy resistance through impeding antigen presentation to T cells [19,55,60,61,62]. In a small proportion of melanomas, loss of HLA class I expression is caused by mutation of the *B2M* gene [60,63]; however, epigenetic mechanisms or translational dysregulation are much more frequent mechanisms of HLA class I loss in cancers [64,65]. Loss of PTEN expression has been implicated as a mechanism of primary and acquired resistance to ICI therapy in melanoma [66,67]. Moreover, a cluster of cancer-germline antigens, located in chromosome Xq28, predicted resistance to CTLA-4 blockade (but not to PD-1 blockade) in melanoma patients [68]. Several studies have identified alterations of genes associated with interferon-γ (IFN-γ) signaling as mechanisms of immunotherapy resistance, such as mutations of *JAK1/2* (Janus kinase 1/2) [63], loss of *IFNGR1/2* (interferon gamma receptor 1/2) and gains of *SOCS1* (suppressor of cytokine signaling 1) and *PIAS4* (protein inhibitor of activated STAT 4) genes [69], as well as mutation of *SERPINB3/4* (serpin family B members 3/4) genes [70], suggesting that IFN signaling plays a crucial role in these processes. Furthermore, an IFN-γ-related mRNA profile was found to be predictive of response to anti-PD-1 therapy in multiple tumors, including melanoma [71]. More recently, a type-I IFN resistance gene expression signature was identified in a human melanoma model, where a 17-gene component was found to be predictive for ICI therapy efficacy [72]. Collectively, there are several genetic alterations, inherent or acquired in melanoma progression, that have been defined to be associated with ICI therapy efficacy, which all wait for prospective clinical validation.

## 10. Concluding Remarks

Molecular pathology plays a critical role in the diagnostics and management of malignant melanoma. Extensive genetic analyses of melanoma patients not only resulted in the proper molecular classification of the tumors but also discovered several genetic conditions which are responsible for the hereditary forms or the increased risk of development. Since the diagnostics of melanoma can be difficult due to a wide range of premalignant melanocytic lesions, molecular markers and gene expression signatures can support diagnostics. Furthermore, the molecular pathology analyses revealed the genetic background of malignant progression and also identified clinically useful gene expression signatures with prognostic value and several potential targets for new therapies. However, the advent of target therapies and immunotherapies of melanoma present another array of selection pressure for the development of novel tumor cell clones characterized by selective genetic advantages for resistance toward those therapies. Accordingly, a continuous monitoring of the genetics of the progressing disease is necessary to optimize clinical management.

## Figures and Tables

**Figure 1 ijms-23-05384-f001:**
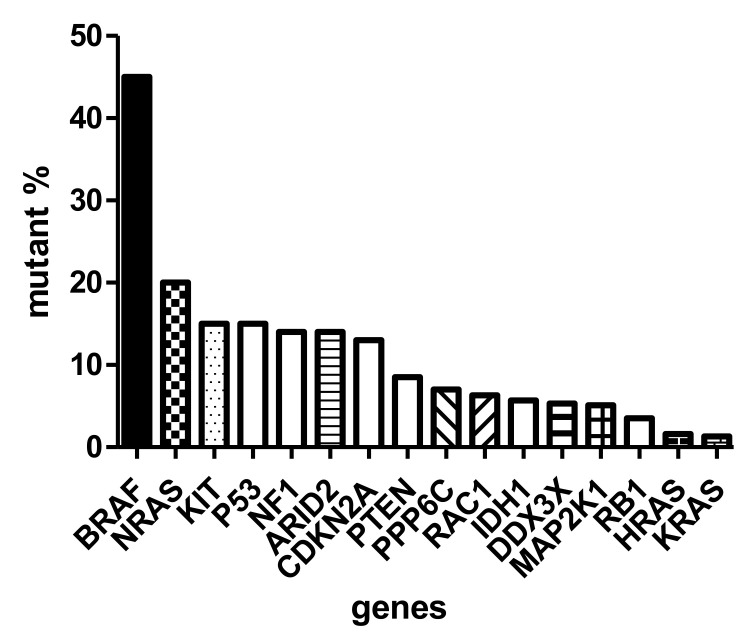
Mutation spectrum of driver genes (oncogenes and suppressor genes) of skin melanoma based on TCGA (The Cancer Genome Atlas).

**Table 1 ijms-23-05384-t001:** Molecular subtypes of melanoma [8].

	*BRAF*-Mutant	*RAS*-Mutant	*NF1*-Mutant	Triple Wild-Type
MAPK Signaling	+	+	+	−
**Cell cycle**	*CDKN2A*mut 60% *CDK4*mut rare	*CDKN2A*mut 70% *CDK4*mut rare CCND1amp 10%	*CDKN2A*mut 70% *RB1*mut 10%	*CDKN2A*mut 40% *CDK4*amp 15% *CCND1*amp 10%
**DDR**	*TP53*mut 10%	*TP53*mut 20%	*TP53*mut 30%	*MDM2*amp 15%
**Epigenetics**	*ARID2*mut 15% *IDH1*mut	*ARID2*mut 15% *IDH1*mut	*ARID2*mut 30% *IDH1*mut	*IDH1*mut
**Others**	*PPP6C*mut 10% *PD-L1*amp *MITF*amp	*PPP6C*mut 15%		

amp, amplification; DDR, DNA damage response; mut, mutation.

**Table 2 ijms-23-05384-t002:** Immunohistochemical markers of melanoma [13].

Marker	Cellular Localization	Sensitivity (%)	Specificity (%)
S100B	cytoplasm	>93	low
Pmel-17/gp100	melanosome	>70	>90
MART-1/MelanA	melanosome	>85	>95
tyrosinase	melanosome	>80	low
MITF	nuclear	>80	low
SOX10	nuclear	>95	low

**Table 3 ijms-23-05384-t003:** Molecular markers of malignancy of melanocytic lesions [16,17].

FISH [16]	myPath [17]	
		**TME**
		*CCL5*
*RREB1*amp	**Tumor**	*CXCL9/10*
*CCND1*amp	*PRAME*	*CD38*
CDKN2A LOH	*S100A7/8/9/12*	*IRF1*
*MYB* LOH	*PI3*	*LCP2*
		*PTPRC*
		*SEL1*

amp, amplification; LOH, loss of heterozygosity; TME, tumor microenvironment.

**Table 4 ijms-23-05384-t004:** DecisionDx prognosticator of melanoma: list of genes to be evaluated [34].

Gene Symbol	Gene Name	Regulation
*BAP1*	BRCA1-associated protein 1	down
*MGP*	Matrix G1a protein	down
*SPP1*	Osteopontin	up
*CXCL14*	Chemokine ligand 14	down
*CLCA2*	Chloride channel accessory 2	down
*S100A8*	S100 Ca-binding protein A8	down
*S100A9*	S100 Ca-binding protein A9	down
*BTG1*	B-cell translocation gene 1	down
*SAP130*	Sin3A-associated protein	down
*ARG1*	Arginase 1	down
*KRT6B*	Keratin 6B	up
*KRT14*	Keratin 14	down
*GJA1*	Gap junction protein A1	down
*ID2*	Inhibitor of DNA binding 2	down
*EIF1B*	Eukaryotic translocation initiator 1B	up
*CRABP1*	Cellular retinoic acid binding protein 1	down
*ROBO1*	Roundabout guidance receptor 1	down
*RBM23*	RNA binding protein 23	down
*TACSTD2*	Tumor-associated Ca-signal transducer 2	down
*DSC1*	Desmocollin 1	down
*SPRR1B*	Small proline-rich protein 1B	down
*TRIM29*	Tripartite motif 29	down
*AQP3*	Aquaporin 3	down
*TYRP1*	Tyrosinase-related protein 1	down
*PPL*	Periplakin	down
*LTA4H*	Leukotriene A4 hydrolase	down
*CST6*	Cystatin E/M	down

## Data Availability

Not applicable.

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
