# Peer review of "Molecular Pathology of Skin Melanoma: Epidemiology, Differential Diagnostics, Prognosis and Therapy Prediction"

_ijms, 2022, doi:10.3390/ijms23105384_

Round 1

Reviewer 1 Report

The manuscript of "Tímár and Ladányi", "Molecular pathology of skin melanoma: epidemiology, differential diagnostics, prognosis and therapy prediction" aims of synthesizing the knowledge of skin melanoma genomics in order to show its clinicopathological relevance and contribute for the translation of this knowledge into clinical practice.
The authors present a comprehensive review of the genetic markers associated with the different types of melanomas and their application in the characterization of prevalence, molecular profile, diagnosis, progression and prognosis, presenting examples of how this knowledge can be applied in the direction of therapy.
The manuscript is a massive source of information and a good document for the reader to learn about the genetic variability associated with melanoma. However, there are some points that, if addressed, can improve the impact of the manuscript:
(i) The authors did not explore how the different genetic signatures could be translated into metabolic alterations that explain the greater resistance to therapies, the greater capacity for metastasis and immunosuppression in the tumor microenvironment. Such an exercise would help to better understand how this knowledge can be transferred to the clinic.
(ii) Sometimes, the text is difficult to read because of punctuation and because of the (inevitable) extensive use of abbreviations. The first can be solved with a more careful review of the text, namely in the placement of commas; the second can be solved by placing a list of abbreviations.
(iii) The points at which the authors believe that translation in clinical practice could be further explored should be presented more assertively. The text does not make it clear at which points the required genetic knowledge already exists to transfer it to the clinic.
(iv) Reading the "Concluding remarks" is almost an anti-climax of what the authors promise in the Introduction. They end with the statement "Accordingly, a continuous monitoring of the genetics of the progressing disease is necessary to optimize clinical management". However, the text lists genetic markers but does not present any solid pattern of genetic markers that characterize the progression of the disease, and to be used to assess the impact of therapy.
 Minor points:
A. Legend of Figure 1 should be improved. Shown are the genes that are most frequently mutated in melanomas or "the driver genes"? What is a "driver gene"? And what is TCGA? 

B. The content of Table 1 hardly has any connection with what is in the text and information included in the table is not discussed in the text. 

C. Line 173: is "(My Path)" or "Mel-Get-14"?

Author Response

Response to Reviewer 1.

  • Critique: Lack of description of the connection of genetic signatures of melanoma to metabolic characteristics.

Response: We agree with the critique and corrected the manuscript accordingly (see lines 280-295, and used three new recent references (Alkaraki et al. 2021, Nath et al 2016, and Valenti et al. 2021)

  • Critique: Text is difficult to read

Response: We have carefully checked punctuation with special attention to the placement of commas. Moreover, we included a (rather long) list of abbreviations. We hope that the text is easier to understand after these corrections. Furthermore, an extended list of abbreviations is provided (L428-468)

  • Critique: Which are those new genetic information which can be translated to clinic? There are only listings but no translational comments. Primary versus metastasis, etc.

Response: We agree with the critique and corrected the relevant paragraph (lines 239-267). We point out the importance of BRCA1 mutation as a potential PARP inhibitor target, we also suggest that the immunogenomic characteristic of lung metastasis exclusively may raise the possibility of having better ICI response as compared to other metastases. On the other hand, the emergence of HGF-MET genetic alterations in brain metastases may offer novel therapeutic options by using MET inhibitors. In case of liver metastases the CDK6 and MAPK amplifications may offer other novel therapeutic options since inhibitors are available.

  • Critique: Concluding remarks

Response: We are sorry if the Conclusion  seemed a bit “downgraded”: we carefully revised the text but still feel that this Conclusion is correct…. However, the Abstract was corrected, since that was not in line with the main text nor with the Conclusion.

Minor A. Critique: Fig. 1 legend should be improved.

Response: Sorry for that. TCGA is explained (a public database of human cancers describing the most important mutated genes, the Cancer Genome Atlas program of NIH/CI). Driver genes: explanation is given as mutated oncogenes and suppressor genes.

Minor B. Critique: Information in Table 1 is not described in the text

Response: Table 1 demonstrates how different melanoma oncogenic mutations associate with each other to form four different molecular genetic classes of melanoma. We have completed with missing issues of epigenetics and PPP6C, see in lines 118-123.

Minor C. Critique: Line 173: MyPath versus Mel.Get-14

Response: This problem concerns the heading of Table 3.  The gene expression signature MyPath from Myrade is composed of 14 genes so it is a 14-gene signature (incorrectly called Mel-GES14). This misleading header is corrected to MyPath which is the trade name, since it is a commercially available test.

Reviewer 2 Report

This review manuscript is well written and very informative.

Related to biomarkers to ICI therapy, I suggest the following papers to be added if possible.

[gene signature]

1) Ayers, M. et al. IFN-gamma-related mRNA profile predicts clinical response to PD-1 blockade. J Clin Invest 127, 2930-2940, doi:10.1172/JCI91190 (2017). It shows that IFN-γ gene signature (IDO1, CXCL10, CXCL9, HLA-DRA, STAT1, IFNG) predicts the response to anti-PD-1 therapy in multiple tumors including melanoma.

2) Shukla, S. A. et al. Cancer-Germline Antigen Expression Discriminates Clinical Outcome to CTLA-4 Blockade. Cell 173, 624-633 e628, doi:10.1016/j.cell.2018.03.026 (2018).

This paper shows that gene signature of cancer germline antigen genes (MAGEA3, CSAG3, CSAG2, MAGEA2, CSAG1, MAGEA12, MAGEA6) is a predictor of resistance against anti-CTLA-4 blockade.

[microbiota]

1) McCulloch JA, Davar D, Rodrigues RR, Badger JH, Fang JR, Cole AM, Balaji AK, Vetizou M, Prescott SM, Fernandes MR, Costa RGF, Yuan W, Salcedo R, Bahadiroglu E, Roy S, DeBlasio RN, Morrison RM, Chauvin JM, Ding Q, Zidi B, Lowin A, Chakka S, Gao W, Pagliano O, Ernst SJ, Rose A, Newman NK, Morgun A, Zarour HM, Trinchieri G, Dzutsev AK. Intestinal microbiota signatures of clinical response and immune-related adverse events in melanoma patients treated with anti-PD-1. Nat Med. 2022 Mar;28(3):545-556. doi: 10.1038/s41591-022-01698-2. Epub 2022 Feb 28. PMID: 35228752. 

Author Response

We are greatful for the reviewer2 for the critique. We included the 3 suggested references into the corrected version.